# Health Education Programmes to Improve Foot Self-Care Knowledge and Behaviour among Older People with End-Stage Kidney Disease (ESKD) Receiving Haemodialysis (A Systematic Review)

**DOI:** 10.3390/healthcare10061143

**Published:** 2022-06-20

**Authors:** Layla Alshammari, Peter O’Halloran, Oonagh McSorley, Julie Doherty, Helen Noble

**Affiliations:** 1Medical Biology Centre, School of Nursing and Midwifery, Queen’s University Belfast, 97 Lisburn Rd, Belfast BT9 7BL, UK; p.ohalloran@qub.ac.uk (P.O.); o.mcsorley@qub.ac.uk (O.M.); julie.doherty@qub.ac.uk (J.D.); helen.noble@qub.ac.uk (H.N.); 2College of Nursing, University of Hail, Hail 2440, Saudi Arabia

**Keywords:** end-stage kidney disease, haemodialysis, foot ulceration, foot care education, systematic review

## Abstract

Background: ESKD is a total or near-permanent failure in renal function. It is irreversible, progressive and ultimately fatal without peritoneal dialysis (PD), haemodialysis (HD) or kidney transplantation. Dialysis treatments can create new and additional problems for patients, one of which is foot amputation, as a result of non-healing wounds and vascular complications. The association between dialysis therapy and foot ulceration is linked to several factors: physical and psychological health; peripheral arterial disease (PAD); mobility; tissue oxygenation; manual dexterity; neuropathy; visual acuity; anaemia; nutrition; leg oedema; hypoalbuminemia; infection; inadequacy of dialysis; and leg/foot support during dialysis. The potential risk factors for foot ulceration may include: not routinely receiving foot care education; incorrect use of footwear; diabetes duration; neuropathy; and peripheral arterial disease. Aim: The aim of this review is to examine the factors that help or hinder successful implementation of foot care education programmes for ESKD patients receiving haemodialysis. Method: A comprehensive literature search was completed using five electronic databases. Medline; CINAHL; Embase; PsycINFO; and Cochrane Library. The Joanna Briggs Institute checklist (JBI) was used to quality appraise full text papers included in the review. The systematic review was not limited to specific categories of interventions to enable optimal comparison between interventions and provide a comprehensive overview of the evidence in this important field of foot care. Results: We found no previously published studies that considered foot care education programmes for haemodialysis patients who are not diabetic; thus, the present systematic review examined four studies on diabetic patients receiving haemodialysis exposed to foot care education programmes from various types of intervention designs. Conclusions: This systematic review has provided evidence that it is possible to influence foot care knowledge and self-care behaviours in both diabetic patients receiving haemodialysis and healthcare professionals.

## 1. Introduction

End-stage kidney disease (ESKD) is a worldwide public health issue characterised by significant impairment of kidney function [1]. Foot problems and complications, such as foot ulceration (FU), are becoming more prevalent in patients with ESKD, and dialysis treatments are independent risk factors for foot ulceration [2]. Preventing foot complications in ESKD patients who are receiving haemodialysis is essential to minimise the risk of foot ulceration, reduce mortality and enhance patients’ quality of life [3]. Therefore, foot care assessment, intensive foot care education and referral to a foot care specialist clinic for those at risk of foot ulceration may prevent mortality and morbidity in this population [4]. Foot health (foot self-care) is often administered through professional foot care services or by patients who provide their own foot care. It is critical for patients with ESKD receiving haemodialysis to achieve desired outcomes, such as lower incidence of foot ulceration, generally through an educational approach [5].

## 2. Background

ESKD is a total or near-permanent failure in renal function. It is irreversible, progressive and ultimately fatal without peritoneal dialysis (PD), haemodialysis (HD) or kidney transplantation [6]. From 2003 to 2016, global prevalence of ESKD per million population (PMP) grew consistently, with the biggest proportional increases occurring in low- and middle-income countries. Although dialysis is a life-saving procedure, it is also extremely costly. Therefore, its use is restricted in low-income nations with inadequate healthcare resources. Dialysis was prevalent in 1176 PMP in higher-income countries in 2010, 688 PMP in upper-middle-income countries and 170 PMP in lower-income countries. However, dialysis is the most frequent kind of kidney replacement therapy globally, comprising 78 percent of all therapies. Overall, 11 percent of dialysis patients receive peritoneal dialysis and the rest haemodialysis [7]. Dialysis was used by 2.62 million individuals worldwide in 2010, and the need for dialysis is expected to quadruple by 2030 [8].

However, dialysis treatments can create new and additional problems for patients, one of which is foot amputation, as a result of non-healing wounds and vascular complications [2]. Amputation in patients with dialysis is associated with a four-fold increase in the risk of mortality in patients without diabetes (hazard ratio (HR) 4.6 (95% CI 2.8–7.6)), and a similar risk factor exists in patients with diabetes (HR 4.6 (95% CI 3.3–6.4)). This suggests that mortality risk is similar in both diabetic and non-diabetic patients [9]. Prevalence of risk factors for FU is high in ESKD patients receiving haemodialysis, similar to those with diabetes [10]. The association between dialysis therapy and foot ulceration is linked to several factors: physical and psychological health; peripheral arterial disease (PAD); mobility; tissue oxygenation; manual dexterity; neuropathy; visual acuity; anaemia; nutrition; leg oedema; hypoalbuminemia; infection; inadequacy of dialysis; and leg/foot support during dialysis [2].

In one observational study, dialysis nurses implemented a routine foot check in patients with diabetes on haemodialysis in the haemodialysis unit as part of standard clinic care at Fresenius Medical Care North America clinics and reported an association between prevalent foot ulceration and haemodialysis therapy [3]. The potential risk factors for foot ulceration may include: not routinely receiving foot care education; incorrect use of footwear; diabetes duration; neuropathy; and peripheral arterial disease [3]. Between October 2006 and March 2008, in a cross-sectional study of dialysis units in Manchester Royal Infirmary, [2] concluded that dialysis treatment was related independently to foot ulceration and amputation, and that dialysis was a significant risk factor in foot ulceration, which could be prevented with intensive foot care. Patients treated with dialysis may not receive appropriate foot care because healthcare practitioners in dialysis units may be too concerned with the technological demands of dialysis, such as checking and recording weight and vital signs, ensuring that haemodialysis treatments are administered properly, monitoring patients during treatment to detect negative reactions, preparing nursing care plans and working with dialysis technicians to ensure that equipment and dialysis machines are functioning properly. Foot care interventions, including education, and scheduling foot evaluations during or immediately after dialysis treatment may be key elements in successfully preventing foot complications [2].

Patient education plays a vital role in nursing practice and can impact patient health and quality of life positively [11]. Furthermore, nurses play an important role in preventing lower-extremity amputation and foot ulceration through educational interventions, providing foot health care and screening those at high risk [12]. It is important that nurses teach patients how to care for their feet, e.g., performing a physical examination of their feet daily to prevent lower limb complications [13]. For example, nurses can encourage patients to perform a sequence of simple routines to help prevent foot ulcers or ulcer recurrence, such as keeping the feet clean, practising proper skin and nail care, checking shoes before wearing them and choosing the right shoes [14]. A positive relationship between patients and nurses can serve as a foundation for self-care practices related to foot care and can be integrated into dialysis facilities as routine practice [3].

Multiple studies suggest that patient education regarding foot care is effective in preventing foot ulcers and amputation in diabetic patients [15,16,17,18,19]. However, few studies have examined foot ulceration and lower limb amputation (LLA) prevention programmes in patients receiving dialysis [20].

To sum up, foot ulceration, amputation and other serious complications necessitate development of educational interventions to increase knowledge and shape behaviours in ESKD patients concerning foot ulceration risk factors, as well as promotion of self-care knowledge and positive foot self-care behaviours. The goal of educational interventions is to help ESKD patients receiving haemodialysis change behaviours to reduce the incidence of foot ulceration and amputation [21]. Podiatry care advice (e.g., foot skin care, the correct way to wash the feet, proper choice of socks and shoes, and intensive foot examination and patient education) involving individualised patient education on potential foot complications can reduce foot ulceration and reduce the need for amputation in diabetic patients receiving haemodialysis [19]. Intensive patient education programmes are inexpensive and easy to implement in all hospital settings.

Foot care education programmes for diabetic patients are common, as diabetic foot ulceration is one of the major complications of diabetes. However, less is known about foot care education programmes for ESKD patients receiving haemodialysis despite the fact that patients with DM and without ESKD have prevalence rates of risk factors similar to those in ESKD patients without DM [22]. Furthermore, no published systematic reviews have examined foot education programmes for ESKD patients receiving haemodialysis. Therefore, related articles were reviewed systematically to examine the factors that help or hinder successful implementation of foot care education programmes for ESKD patients receiving haemodialysis, describe the structure and delivery of foot care programmes, assess the efficacy of educational interventions on foot care knowledge and foot care practice, and identify factors that act as barriers to or facilitators of implementation.

## 3. Materials and Methods

### 3.1. Search Strategy

The search strategy was not limited to a set time frame. A systematic review search strategy was completed between December 2019 and May 2022 with the assistance of a university librarian. Studies were identified after a search of the following electronic databases: Medline; CINAHL; Embase; PsycINFO; and Cochrane Library. The search focussed on advanced searches using Boolean logic with the logical operators ‘AND’ and ‘OR’ (see Table 1). Google Scholar also was searched for grey literature. The search strategy also included reference lists contained within review studies and other relevant published reviews. The search methods completed for the review are presented in Table A1 in Appendix A. All databases were searched individually, and a combined search was completed subsequently. Endnote X9 was used to remove duplicate studies and manage references electronically. Review studies’ quality was evaluated for inclusion using the Joanna Briggs Institute’s (JBI) critical assessment methods [23].

### 3.2. Eligibility Criteria

This review included studies with the following characteristics:Study participants who were aged over 18 years, diagnosed with ESKD and receiving dialysis.Education programmes in relation to preventing foot ulceration/amputation in patients receiving dialysis.Education programmes conducted by healthcare staff (e.g., dialysis nurses, health educators and podiatrists).Randomised control trials (RCTs), quasi-randomised controlled trials, experimental study designs (e.g., non-RCTs and quasi-experimental, pre- and post-test studies) and prospective observational studies for inclusion (but not case reports).Published in English (because of limited resources for language translation).No criteria on year of publication were employed.Only primary studies were included; researcher opinion papers, reviews, dissertations, editorials and conference abstracts were excluded.

### 3.3. Selection of Studies

The systematic review protocol was based on the Preferred Reporting Items for Systematic Reviews and Meta-Analyses (PRISMA) guidelines and flowchart [24] (see Figure 1).

During the identification phase, 504 publications were identified, of which 110 were duplicates and were removed, leaving a total of 394 titles and abstracts from all databases that four authors (L.A., H.N., P.O. and O.M.) initially screened independently for eligibility using inclusion and exclusion criteria. After initial screening, the review team read 15 full-text articles and discussed the content to assess suitability for inclusion and resolve any disagreements concerning inclusion or exclusion. At this stage, 11 studies were excluded, as they did not meet the eligibility criteria; thus, four studies ultimately were included in the final review.

The final phase of the process involved appraising the research quality of the four studies selected for inclusion.

### 3.4. Methodological Quality Assessment Tool

The JBI’s Critical Appraisal Checklist for Randomised Controlled Trials and Non-Randomised Controlled Studies (Non-Experimental Studies and Quasi-Experimental Studies) was used to evaluate all included studies critically [23]. JBI is designed to be used in systematic reviews to evaluate each study’s methodological quality and determine the extent to which a study has addressed rigour, bias, study conduct and analysis of results. The JBI’s critical appraisal tools include 8–13 questions (depending on study design) used to analyse systematic reviews or meta-analyses. Each item is scored ‘yes’, ‘no’, ‘unclear’ or ‘NA’ (not applicable).

The JBI’s Critical Appraisal Tool for Analytical Quasi-Experimental Studies was used to screen the final list of quasi-experimental studies across different domains, including cause and effect, similarity of participants, similarity of treatment received other than the intervention being studied, control groups, pre- and post-intervention measurements, follow-up, consistency of participant measurement, reliability of outcome measurement and appropriateness of statistical analyses performed (see Table A2 in Appendix A).

The review team assigned specific ratings to each of the included studies and evaluated each study as low, moderate or high quality. Rating scores were calculated by collecting the number of ‘yes’ responses to all individual criteria. Overall quality was measured by aggregating the number of ‘yes’ answers in all individual criteria. Depending on the criteria, studies could receive a score of up to 9. A score greater than 5 indicated a high-quality study, a score between 3 and 5 indicated a moderate-quality study and a score lower than 3 suggested a low-quality study.

Using the appropriate JBI tool, the first named author (L.A.) implemented the first quality assessment of all included studies, which the review team (L.A., H.N., P.O. and O.M.) then reviewed.

### 3.5. Data Extraction

Data extraction is a method that describes the collection of necessary information about study characteristics and findings from studies included in a systematic review [25]. In this review, L.A. conducted data extraction on all included studies, then provided a summary using a data extraction sheet. Data extracted included: author name; year of publication; study title; study country and setting; study aim; design; participants and sample size; characteristics of educational interventions; duration and follow-up strategies; outcomes; and findings. L.A. extracted these data independently (see Table 2), and H.N., P.O. and O.M. cross-checked extraction accuracy to ensure completeness and accuracy.

### 3.6. Data Synthesis and Analysis

A narrative synthesis of research findings was conducted due to the diversity of the chosen methodology, interventions, topics, data heterogeneity and method of reporting outcomes.

## 4. Results

### 4.1. Search Results

Altogether, 504 papers were identified from five databases. After 110 duplicates were removed, the remaining 394 studies were screened based on titles and abstracts, of which 379 were excluded, and then the remaining 15 were subjected to a full-text review. Another 11 did not meet the inclusion criteria after the full-text screening. Of these, one focussed on diabetic renal transplant patients [30], and four did not deal with foot care education interventions [31,32,33,34], one study focussed on improving education and care management of diabetes to improve glycaemic control, alter patient behaviour and reduce complications in the setting of the dialysis unit [35]. Another study focussed not on foot care education but on diabetes patient education for glycaemic control, lipids, blood pressure, diabetes and complication screening [36], two were conference abstracts [37,38] and two were protocol papers [39,40]. Thus, the remaining four papers were included in the review [26,27,28,29]. An overview of the study selection process is presented in Figure 1.

### 4.2. Study Characteristics

Table 2 provides a summary of study characteristics, comprising research aim, design, setting, participants, instruments, follow-up strategies and methodological quality assessment in articles included in this review.

#### 4.2.1. Study Aim, Design, Setting

Two studies were conducted in the United Kingdom [28,29], one in the United States [27] and one in Canada [26]. Brand et al. [28] evaluated the effects of a nurse education programme on patient-reported foot checks and foot care behaviour. The authors published an additional paper in 2016 [29], focussing on improving foot care among diabetes patients receiving haemodialysis. Other health professionals were involved in this study, but the same patients were involved in both studies. Neil et al. [27] focussed on preventing foot ulcers in patients with diabetes and ESKD. Reda et al. [26] examined the effect of a foot care education programme designed to prevent lower-extremity complications in diabetic ESKD patients. Two studies used a quasi-experimental design [26,27], and the other two used a non-randomised stepped-wedge design [28,29]. All studies were published between 2003 and 2016.

#### 4.2.2. Study Participants

The study participants were mainly patients, and the patient groups studied were those with diabetes and receiving haemodialysis [26,27,28,29]. All four studies included older adults, and two included healthcare professionals as respondents [28,29]. Each study’s participant population is summarised in Table 2.

#### 4.2.3. Study Instruments Used to Measure Outcomes

Instruments were used to measure outcomes across the studies reviewed in this article to evaluate knowledge behavioural outcomes following foot care education interventions. Most researchers used a questionnaire as their research tool. To evaluate behavioural outcomes, two studies used the Nottingham Assessment of Functional Footcare (NAFF) Questionnaire [28,29]. The Nottingham Assessment of Functional Footcare (NAFF) was developed to evaluate diabetics′ foot care behaviours [41]. To assess patient foot care knowledge outcomes, one study used the Siriraj Foot-care Questionnaire [42].

#### 4.2.4. Study Follow-Up

From the existing literature, the duration of educational interventions and follow-ups ranged from four to eight months. Intervention evaluations were conducted every two months [28,29], four months [26] or six months [27].

#### 4.2.5. Study Methodological Quality Assessment

Four authors (L.A., H.N., P.O. and O.M.) assessed methodological quality independently using the appropriate JBI checklist, and assessment discrepancies were resolved through joint discussions until consensus was reached. If study sections were judged to be unclear (U) or not applicable (NA), they were assigned zero points, with total scores for each study on their corresponding scales tabulated. The scales used to grade papers assessed methodological processes, including the quality of literature search strategies, data presentation and use of consecutive participant inclusion. While this approach does not provide a comprehensive critical appraisal of a paper, it enables standard comparisons between papers. Each reviewed study’s methodological quality is summarised in (Table A2 in Appendix A). The studies in this review were judged to be of moderate to high quality with a low risk of bias. Table A2 in Appendix A shows that three studies were identified as high quality [27,28,29] and one as moderate quality [26].

### 4.3. Programme Characteristics

The educational interventions’ programme structure, delivery, content and implementation were analysed.

#### 4.3.1. Programme Structure and Delivery Characteristics

One study used foot care education programmes as the intervention, comprising multiple education sessions (group and/or individual) of varying frequency and duration [27] (see Table 3). Education delivery took the form of individual training (*n* = 2) [26,27]; group training (*n* = 1) [27]; paper handouts (*n* = 1) [26]; and/or a lecture format (*n* = 3) [27,28,29]. A delivery format using adjusted provision footwear (shoes) was used in one study [27], provision of foot measurement was used in two studies [26,27] and podiatry care was used in one study to promote foot health [26]. Foot care examination/assessment was undertaken in all four studies [26,27,28,29]. In [27], the authors used a lower-extremity exam, neural exam (10 g Semmes–Weinstein monofilament scale) and vascular exam (toe pressure) (plethysmograph scale) to assess each patient’s foot. Several authors used a standardised physical examination of the lower extremities [26,28,29].

#### 4.3.2. Programme Content and Implementation Characteristics

All four studies focussed on assessing patients’ feet and providing foot care information to patients. The procedures for the educational interventions in each reviewed study are summarised in Table 4.

Brand et al. [29] showed on an educational intervention for nurses, provided sequentially at four dialysis units in which an experienced diabetes podiatrist recruited experienced nurses within dialysis units. The programme advised recruited nurses to conduct a foot examination procedure on all diabetes patients on a monthly basis. It also sought clarification of referral processes to specialist services and foot care information for patients.

Brand et al. [28] showed on an experienced diabetes podiatrist who provided a single education session to six identified nurses that included monthly foot examination, clarification of referral processes to specialist services and foot care information for patients. The nurses who were trained were diabetes link nurses in each of the respective units. These nurses had an interest in diabetes care, and training was congruent with their link nurse roles. Nurses who received training were encouraged to pass this information on to their colleagues. The foot check aimed to identify active, previously unreported foot problems and to deliver foot care advice.

Reda et al. [26] showed on registered nurses with training in foot and wound care who routinely assessed patients while they were receiving haemodialysis over a four-month period, including lower extremities after patients’ shoes and socks were removed, delivering standardised instructions about foot care and footwear to diabetic ESKD patients receiving haemodialysis [26]. Patients were asked to wear appropriate protective footwear as much as possible, check their shoes and socks for potential pressure points, keep their feet adequately moisturised, watch for possible development of calluses and ulcers, and maintain a healthy lifestyle, including smoking cessation and striving for normoglycaemia. On a daily basis, nails were clipped and calluses pared. Consultants in orthopaedic and vascular surgery, infectious diseases and wound care assessed and treated any ulcers found. Adjunctive measures—including orthoses, custom-moulded insoles and properly fitting orthopaedic footwear—were prescribed as needed [26] (see Table 4).

Neil et al. [27] showed on a foot care education programme that included a four-part intervention comprising foot assessment, individual foot care education, group foot care education and the use of special shoes in the experimental group [27]. The first part, comprising the foot assessment, used visual inspection to detect muscle wasting, the presence of foot ulcers, foot deformities and skin conditions and used the Semmes–Weinstein 5.07 (10 g) Monofilament Examination (SWME) for sensory testing [27]. The SWME is a simple, cheap and reproducible method for evaluating the sensorial component of neuropathy—i.e., the absence of protective sensation (loss of protective sensation)—comprising the Semmes–Weinstein monofilament kit, which has been recommended consistently as a screening tool for identifying diabetic patients at risk of ulceration and amputation [43,44,45,46]. The SWME has been identified as the best available screening instrument for neuropathy [45] (see Table 2). During the vascular exam, the Pulse Oximeter Pleth (plethysmograph) scale was used to determine toe blood pressure. To assess the probability of healing lower-extremity ulcerations, this test is performed on the big toe. A pressure of 30–40 mmHg indicates a good possibility of healing. During the second and third parts of the study, in terms of individual and group foot care education, patients received care education sessions, including printed handouts, from a certified diabetes educator. The foot care education sessions’ content included foot inspection, foot cleaning, cleaning agents, management of dirty feet, cleaning between toes, drying between toes, toenail cutting and the perils of going barefoot. The control group in this study did not receive any foot care education (see Table 4). The final part addressed within this study was foot shoe size. Nail et al. [27] used a standard Brannock Device, the most widely used foot-measuring device among footwear companies in the design of shoes [47], to measure the foot’s length, arch length and width [48].

### 4.4. Foot Care Education Efficacy

Educational foot care interventions generally elicited positive outcomes (see Table 4). The effects from interventions can be categorised under the following headings: foot care knowledge; foot care behaviours; foot examination; lower limb function; and referral rates.

#### 4.4.1. Programme’s Effect on Patients’ Foot Care Knowledge

In terms of foot care knowledge outcomes, one study assessed the foot education programme’s impact on patients’ foot care knowledge [27]. This study demonstrated that foot assessment and education elicited better patient knowledge about foot care. In this study, the authors used an experimental group and a control group. The control group registered lower scores on the foot care knowledge post-test overall, with nine doing worse and seven better. However, in the experimental group, three did worse, five did better and two received the same score.

#### 4.4.2. Programme’s Effect on Patients’ Foot Care Behaviour

All four studies highlighted the effect of patient education on foot self-care behaviours.

Neil et al. [27] used a convenience sample of 24 adult men and women with diabetes, receiving haemodialysis in a quasi-experimental pilot study. The study demonstrated higher scores on foot care behaviour in patients following educational interventions for the experimental group; however, the control group registered lower scores on the post-test on foot care behaviour. The experimental group scored higher on a post-test of their foot care behaviour. For example, 80% of the experimental group would clean their feet right away after the intervention, compared to 44% in the control group. This led the authors to conclude that nephrology nurses need to be more vigilant with their clients’ foot care. Reda et al. [26] found that a greater proportion of current patients than previous ones were using adequate footwear after the intervention, which is important because inappropriate footwear may contribute to the development of diabetic foot ulcers. Brand et al. [28] reported that a single education session improved self-reported foot care behaviour, reflecting greater awareness of risks among diabetes patients receiving haemodialysis. The results also indicated a significant difference in behaviour between the baseline and after eight months in the sample based on the NAFF Questionnaire results. Brand et al. [29] found a notable increase in NAFF scores over time, indicating improvements in patient-reported self-care behaviours. However, most changes occurred between baseline and second assessments in all units, despite interventions having been implemented at only a single site.

#### 4.4.3. Nurse Education’s Effect on Patients’ Foot Examination

Two studies highlighted the effects from foot examinations by nurses, with change occurring following the intervention programme, suggesting that education provided by nurses encouraged study participants to conduct more regular foot examinations [28,29]. When lectures were the only educational delivery method, frequency of foot examinations in both patients and healthcare professionals improved [28,29]. Moreover, patient interventions were effective when they comprised multiple educational approaches, such as individual training [26,27,28,29] or small-group discussions [27]. Moreover, individual and group training combined with educational handouts [27] improved diabetic foot examinations.

#### 4.4.4. Lower Limb Function

The four studies in this review reported that lower limb problems improved following educational interventions [26,27,28,29]. Studies employing a four-months-or-longer follow-up time offered promising findings regarding stability in improvements (see Table 4), particularly in relation to a reduction in the frequency of peripheral neuropathy [26], the absence of dorsalis pedis and posterior tibial arterial pulses [26], and no new foot ulcer development [27]. Reda et al. [26] supported the hypothesis that foot care programmes comprising nursing assessments and patient education may be associated with a reduction in some diabetic foot complications in ESKD patients. However, in one study, no effect was detected on incidence of ulcers, limb amputation or Charcot foot [26].

#### 4.4.5. Referral Rates

Foot care referral pathways to podiatry increased in diabetic patients receiving haemodialysis [28,29] (see Table 4). No referrals were found directly from haemodialysis nurses to podiatry services for diabetic foot issues prior to foot checks. However, podiatrists referred 18 patients (26 different referrals) to the foot protection team from the two haemodialysis units in the two years since foot checks began. More importantly, none of these foot ulcer cases resulted in any form of hospitalisation or amputation [28,29]. Nurses anecdotally reported that communications between dialysis units and podiatry services improved, and podiatrists also reported that the number of relevant referrals to podiatry services increased, with improved communications between dialysis units and podiatry services [28,29].

To sum up, foot care education programmes reported in previous research indicated improvements in foot self-care knowledge, foot self-care behaviours, foot examination, rates of lower limb problems and referral rates in older ESKD patients receiving haemodialysis and haemodialysis nursing staff.

### 4.5. Programme Implementation

#### 4.5.1. Implementation Barriers

Four studies found several barriers that affected implementation of educational interventions in foot care programmes. These barriers can be grouped into three categories: data-related barriers; organisational barriers; and patient-level barriers.

Commonly reported data-related barriers included the risk of bias due to small sample size [26,27] or unknown response rate [28], as well as partially completed questionnaires at every time point [28,29].

In relation to organisational-level barriers, [28,29] reported non-cooperation between podiatrists and doctors regarding lower-extremity exams, even though they were aware that this was a potential barrier to effective foot care. Moreover, an important factor that mediated the programmes’ impact and sustainability at the organisational level was that some patients could not implement recommendations on footwear and orthoses (shoe inserts) because of a lack of resources to purchase medical-grade footwear [26]. However, different state-based programmes are available in Australia that can offer financial support to diabetes patients who need medical footwear [49].

Barriers at the patient level are due to some patients misunderstanding education in relation to nail cutting and the use of dangerous tools, such as knives or razor blades [27]. Furthermore, most participants (70% in the experimental group and 66.7% in the control group) cut toenails straight across, which served as a barrier to the delivery of effective foot care. A lack of cooperation by patients and limited understanding of the neural test were also identified as potential barriers to programme implementation and subsequent compassionate care delivery [27]. The general health effects of poor foot care functioned as barriers to providing effective care and made foot inspection difficult, e.g., retinopathy and a lack of mobility and flexibility. The time burden associated with haemodialysis resulted in patients experiencing difficulties accessing appropriate specialist services, and a lack of access to services negatively affected intervention outcomes [28,29]. Limited time was reported as a barrier across all foot care programme implementation initiatives.

#### 4.5.2. Implementation Facilitators

Four studies identified potentially beneficial practices in the implementation of foot care interventions. The perception of participants (staff nurses) was the most important facilitator and education sessions and support from advocates were the most common facilitators (Table 4). Participants agreed that the following features also facilitated foot care education programme implementation: teamwork and collaboration; specific resources, such as the presence of an experienced diabetes podiatrist; and availability of supplies, e.g., a neural exam scale (10 g Semmes–Weinstein monofilament scale), vascular exam scale (plethysmograph) and foot measurement scale (Brannock Device) [26,27]. Furthermore, nurses who participated reported that patients’ willingness to ask questions, good communication between patients and nurses, and good communication between dialysis units and podiatry services facilitated programme implementation and subsequent foot care delivery [28,29]. Reda et al. [26] and Neil et al. [27] found that nurse awareness of the importance of foot care education, foot assessment and special shoes positively influenced programme sustainability.

## 5. Discussion

### 5.1. Summary of Evidence

In this systematic review, we searched the literature for publications on interventions to describe foot care programme structure and delivery for ESKD patients receiving haemodialysis. No limitation on the year of publication was applied, as the purpose was to identify all relevant foot-related interventions or study designs (except for case reports). The systematic review was not limited to specific categories of interventions to enable optimal comparison between interventions and provide a comprehensive overview of the evidence in this important field of foot care. This study aimed to examine the factors that help or hinder successful implementation of foot care education programmes in ESKD patients receiving haemodialysis. However, we found no previously published studies that considered foot care education programmes for haemodialysis patients who are not diabetic; thus, the present systematic review examined four studies on diabetic patients receiving haemodialysis exposed to foot care education programmes from various types of intervention designs. Two review studies used a quasi-experimental design [26,27], and two used a non-randomised stepped-wedge design [28,29]. The studies’ methodological quality varied, with three studies identified as high quality and one as moderate quality. The studies were not combined in a meta-analysis because of the high heterogeneity of outcome assessment tools and education interventions used across the included papers. Consequently, this review’s findings were presented in a study-by-study narrative form.

The studies focussed on improving patient foot care knowledge and foot self-care behaviours, as well as on foot care educational interventions, including a complex intervention in which both patients and health professionals (nurses, podiatrists and doctors) were educated. Regarding foot examination, two studies examined healthcare professionals, with one focussing on a nurse, podiatrist and doctor [28] and the other focussing on nurses [29]. Both were effective in improving the rate of foot examinations by nurses only [29].

All these studies focussed on diabetic patients receiving haemodialysis, which is understandable because diabetic foot syndrome is one of the common complications of diabetes mellitus (DM). However, other conditions that affect the feet also require healthcare interventions. For example, adult ESKD patients receiving haemodialysis and who are not diabetic are at high risk for serious foot complications, including foot ulceration and foot amputation, and the risks of ulceration and amputation are similar to those with diabetes [1]. ESKD is linked to a considerable rise in the number of diabetic foot lesions. All foot issues, such as ulceration, infection, gangrene and amputation, fall under this category. The same rules apply to preventing, treating and taking care of diabetic feet in people with ESRD as they do to diabetic feet in general [50].

The educational interventions were complex interventions with multiple components. The structure of patient education can take many forms and use various methods (e.g., individual or group sessions), different intervals (e.g., single session or meetings held every two months), varying lengths of treatment and different educators (e.g., nurses, podiatrists and doctors). All foot care education programmes were provided by personnel trained in the research field who had health and medical science backgrounds (e.g., nurses, diabetic educators and diabetes podiatrists). The benefit of this approach is the provision of quality information to older patients [51]. Four studies used more than one teaching technique, i.e., verbal (e.g., teaching, discussion, assessment) and written (e.g., handouts). The value of such a foot care programme is that it will allow older diabetes patients receiving haemodialysis and their healthcare providers to implement the most efficient approach to encourage foot self-care.

The instruments that have been used to measure outcomes varied across the studies reviewed in this article. Several instruments were used to evaluate knowledge and behavioural outcomes following foot care education interventions. Most researchers used a questionnaire as their research tool. To evaluate behavioural outcomes, two studies used the NAFF Questionnaire [28,29], a 29-item self-reported indicator of the extent to which individuals comply with prescribed foot care behaviours [52]. It was developed by translating the data from printed leaflets available in podiatry departments and hospitals in Nottingham and Derby into a question format. It was designed to recognise patients who were not participating in the recommended practice of foot care and to use as an outcome indicator for educational trials to prevent recurrence in individuals with healed foot ulcers [52]. NAFF has been validated and used in several published studies e.g., [53,54,55,56]. To assess patient foot care knowledge outcomes, one study used the Siriraj Foot-care Questionnaire [42], which was validated and used in previous studies, such as [42].

To prevent foot complications, such as ulceration and amputation, constant care and observation of potential changes in foot health are required by both patients and healthcare professionals [26,27,28,29]. To be able to detect these changes and administer care, foot health knowledge and practice need to be up-to-date. Two studies highlighted patient education’s effect on lower limb function, proper foot self-care behaviours that can reduce the frequency of peripheral neuropathy, absence of dorsalis pedis and posterior tibial arterial pulses, enhancing the adequacy of footwear in someone with an at-risk foot [26] and no new foot ulcer development [27]. However, no significant differences were found in incidence of foot problems after educational intervention, such as amputation, ulcer and Charcot foot. Two studies highlighted patient education’s effect on foot self-care knowledge and its importance in health outcomes. Ideal foot self-care behaviours include daily foot and shoe checks, proper daily foot hygiene, not walking barefoot, wearing appropriate shoe gear, trimming toenails, not using anything abrasive on the feet and routine foot exams by a professional trained to identify foot complications [26,27]. Foot-health knowledge and behaviours need to be updated to detect changes and provide appropriate care.

This systematic review identified several foot health interventions that are recommended for use and development, but to ensure that studies are conducted at a high methodological level, the use of rigorous scientific methods and validated instruments is encouraged strongly. Furthermore, patients’ capacity and understanding of health education also may affect the success and development of foot health interventions. However, in all these studies, educational intervention on foot self-care behaviour was not included for caregivers if their patients suffer, e.g., from retinopathy with a visual disability. Li et al. [57] stated that educational intervention can facilitate positive foot self-care behaviours among retinopathy patients with visual disabilities and their primary caregivers. Not only can this encourage primary caregivers to participate in the management of foot care in terms of supervision and support, but it also can alleviate pressure on patients in the process of foot self-care effectively, helping to establish healthy behaviours [57]. In developing future research, patients’ primary caregivers need to be educated.

To our knowledge, this is the first review to identify factors that act as barriers to or facilitators of implementation of foot care education programmes for patients with ESKD receiving haemodialysis. Four studies found several barriers that affected implementation of educational interventions in foot care programmes. These barriers can be grouped into three categories: data-related barriers; organisational barriers; and patient-level barriers. Commonly reported data-related barriers included the risk of bias due to small sample size [26,27] or unknown response rate [28], as well as partially completed questionnaires at every time point [28,29]. Because of the limited sample size, determining whether a given result is a true finding may be challenging. The second barrier, on the organisational level, was that some patients could not implement recommendations about footwear and orthoses (shoe inserts) because of a lack of resources to purchase medical-grade footwear [26]. However, different state-based programmes are available in Australia that can offer financial support to diabetes patients who need medical footwear [49]. Kooij et al. [58] stated that instead of organising health care around professionals and institutions, some contend that it should be arranged increasingly around patients. Furthermore, on the organisational level, there was a lack of cooperation from podiatrists and doctors regarding lower-extremity exams, even though they were aware that this was a potential barrier to effective foot care programme implementation. A previous study confirmed health professionals’ involvement in all aspects of integrated care delivery and how changes to the health workforce affect implementation of integrated care profoundly [59]. Moreover, poor interprofessional collaboration affects not only intervention implementation but also delivery of health services and patient care [60]. To provide quality care to patients, healthcare professionals must work together as a team [61]. The third barrier, on the patient level, concerns how some patients misunderstand education in relation to nail cutting and the use of dangerous tools, such as knives or razor blades [27]. Consequently, if the researcher delivered foot skin-care practices through providing practical demonstration and practice, it may help older diabetes patients receiving haemodialysis to implement the most effective approaches to promote foot self-care. According to [62], lower extremity examination, foot care, footwear and toenail clipping are technical skills that must be learned, mastered and assessed satisfactorily. As a result, the foot care training programme should include information on typical and abnormal foot deformities, as well as practical demonstrations and practice in foot care, footwear and toenail trimming using proper equipment [62]. However, in a traditional pedagogical style (presentation), the educational interventions were delivered and implemented in person.

This review identified potentially beneficial practices in the implementation of foot care interventions. The perception of participants (staff nurses) was the most important facilitator and education sessions and support from advocates were the most common facilitators. Among the factors that act as facilitators to implementation, the researcher focussed on nurses, as they are key personnel in healthcare delivery and play a vital role in delivering and organising treatment, avoiding adverse effects and maximising productivity in health services and patient outcomes [63]. Furthermore, nurses’ awareness of the importance of foot care education, foot assessment and special shoes positively influenced programme sustainability. For example, to help avoid foot ulcers or recurrence, nurses should advise patients to conduct a series of simple practices, such as testing shoes before wearing them, keeping feet clean and continuing skin and nail treatment. Training patients on how to choose the right shoes is also important [12]. Furthermore, patients’ willingness to ask questions and good communication between patients and healthcare professionals help facilitate programme implementation and subsequent foot care delivery. One of the most critical elements for enhancing patient satisfaction, compliance and health outcomes is effective interpersonal communication between healthcare professionals and patients [64]. To sum up, health professionals—whether nurses, podiatrists or doctors—need to focus on and improve, if necessary, their communication with patients.

### 5.2. Strengths and Limitations

This is the first systematic review to examine the factors that help or hinder the successful implementation of foot care education programmes for diabetes patients receiving haemodialysis. This systematic literature review is a comprehensive examination of foot self-care knowledge and practice interventions conducted worldwide solely on diabetic patients receiving haemodialysis. This review provides important insights on diabetic patients receiving haemodialysis management and care, an area that had been ignored in the literature and interventions. The studies included within this systematic literature review provide evidence of improved knowledge and behavioural outcomes, and how these outcomes ultimately improve quality of life for diabetic patients receiving haemodialysis. One limitation of this review is that it included only four studies. The second limitation is that the researcher failed to find any previously published studies that considered foot care education programmes for haemodialysis patients who are not diabetic. Furthermore, the systematic review did not include studies that examined caregiver foot care knowledge and practices.

## 6. Conclusions

Foot care knowledge and positive behaviours are required to prevent lower extremity complications (ulceration and amputation) for diabetes patients receiving haemodialysis and can impact health care positively if early assessment and foot care education programmes have been undertaken properly. This systematic review has provided evidence that it is possible to influence foot care knowledge and self-care behaviours in both diabetic patients receiving haemodialysis and healthcare professionals. Foot care education approaches identified in this review can be used to improve patient care. Regular foot screening and care are necessary to reduce the risk of foot ulcerations and lower-extremity amputation among ESKD patients receiving haemodialysis. Most foot care education has been implemented through lectures and individual training, contributing to positive outcomes in foot care knowledge and behaviours. Foot-related education studies will benefit from the use of a more systematic process of development and testing in the future. More research is required to promote effective foot care, particularly in those with ESKD who require haemodialysis and who are not diabetic. In the future, researchers and practitioners must implement a vigorous education programme that focusses on ESKD patients receiving haemodialysis who are not diabetic gaining access to foot self-care among the older population. Furthermore, nephrology nurses play an important role in early detection and appropriate interventions when the risk of foot ulceration is high. Thus, researchers and practitioners need to develop tools for early assessment and foot care education programmes to enable the implementation of preventive strategies for nephrology nurses.

## Figures and Tables

**Figure 1 healthcare-10-01143-f001:**
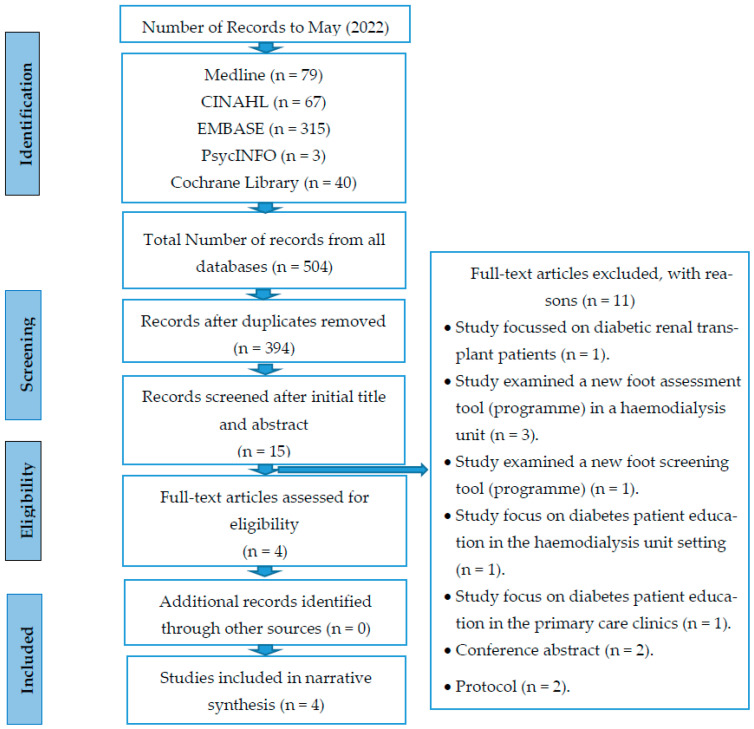
PRISMA 2009 flow diagram for the process of study selection.

**Table 1 healthcare-10-01143-t001:** Search strategies used in electronic databases.

Keywords Used	MEDLINE	CINHAL	Embase	PsycINFO	Cochrane Library
1. (Kidney Failure, Chronic/OR Renal Insufficiency/OR Renal Replacement Therapy/OR Renal Dialysis/OR kidney.mp. OR Kidney/OR renal.mp.)	1,050,412	131,424	1,409,581	9856	75,021
2. (Foot/or foot.mp. OR Diabetic Foot/OR Foot Ulcer/)	122,325	40,744	162,965	7951	13,844
3. (Education/OR Education, Medical, Continuing/OR Education, Nursing, Continuing/OR Education, Professional/OR Education, Nonprofessional/OR Health Education/OR ‘Early Intervention (Education)’/OR Patient Education as Topic/OR Patient Education Handout/OR Education, Nursing/OR education.mp.)	877,716	613,090	1,122,166	601,208	73,585
1, 2 AND 3	84	68	324	3	40
Limits: Language: English	79	67	315	3	40
Total = 504					
After duplication-checking by EndNote: 504 − 110 = 394 articles					
	74	44	247	3	26

**Table 2 healthcare-10-01143-t002:** Descriptions of the studies included in the review.

First Author (Year) Country Title	Study Aim	Study Design	Study Setting	Study Participants E = Experiment C = Control	Instrument(s)	Follow-Up (Measurements)	Quality Appraisal
Reda et al. [26]	(a) To evaluate lower-extremity problems of patients receiving haemodialysis. (b) To compare incidence of complications in patients receiving a foot care education programme and not receiving a foot care education programme.	Quasi-experimental study.	800-bed teaching hospital.	- 58 patients with DM and chronic kidney disease (haemodialysis). - Age: 62 ± 12 years.	- Physical examination of lower extremities. - Neural exam (10 g Semmes–Weinstein monofilament scale). - Toe blood pressure (using a toe cuff and a portable flow Doppler sensor).	Base evaluation after 4–6 months: peripheral neuropathy; decreased or absent peripheral pulses; level of amputation; ulcer; Charcot foot; and footwear status.	6/9
Neil et al. [27]	(a) To describe the self-care practices of patients with diabetes and ESRD on footwear, foot care and behaviour tied to diabetes. (b) To identify whether a reduction in lower-extremity amputations and foot ulcers could be achieved in patients with kidney dialysis through education about foot exams, footwear and foot care. (c) To identify whether lower-extremity cost savings and ulcer prevention would be necessary to cover programme costs.	Quasi-experimental pilot study.	Outpatient haemodialysis treatment facility.	24 patients with diabetes and ESRD E: *n* = 13 C: *n* = 11.	- Siriraj foot care questionnaire. - Lower-extremity exam. - Neural exam (10 g Semmes–Weinstein monofilament scale). - Vascular exam (toe pressure) (plethysmograph scale). - Foot measurement (Brannock Device).	Knowledge questionnaire was administrated pre- and post-intervention and after six months of programme completion.	9/9
Brand et al. [28]	(a) To measure whether a nursing education system increased the frequency with which nurses performed foot checks on diabetics receiving from haemodialysis. (b) To determine effect on self-reported behaviour in foot care.	Non-randomised stepped-wedge design.	Four haemodialysis units in UK.	95 diabetes patients receiving haemodialysis.	- Frequency of foot assessment. - Nottingham Assessment of Functional Foot care (NAFF).	Foot examination frequency was repeated at two-month intervals for eight months.	7/9
Brand et al. [29]	(a) To assess whether training haemodialysis nursing staff to conduct foot examinations and educating patients on foot care influenced the frequency of both foot examinations by nurses and reported foot self-care behaviour by diabetes patients. (b) Self-care behaviour comprises any activities that patients undertook independently of any professional to ensure foot health.	Non-randomised stepped-wedge design.	At four haemodialysis units in the Nottingham area.	95 diabetes patients receiving haemodialysis. Mean age was 67.7 (SD 12.3%), and 52 (54.7%) were men.	- Questionnaire on the frequency of foot assessment by health professionals. - Nottingham Assessment of Functional Foot-care (NAFF) Questionnaire.	The questionnaire was administered to all patients who attended each unit for eight months at two-month intervals. The intervention was introduced sequentially two months apart.	7/9

**Table 3 healthcare-10-01143-t003:** Educational foot health intervention structure and delivery format.

	Author (Year)/Title			
Format of Delivery	Reda et al. [26]	Neil et al. [27]	Brand et al. [28]	Brand et al. [29]
Text message				
Phone calls				
Pamphlet, leaflet, booklet, handout		×		
Lecture		×	×	×
Demonstration				
Hands-on practice				
Audio-visual (Video)				
Individual education	×	×		
Group education		×		
Standard info	×			
Provision of adjusted footwear (shoes)		×		
Provision of insoles or orthoses				
Provision of podiatry care	×			
Provision of foot measurement	×	×		
Foot examination/assessment	×	×	×	×
Application of specific treatment				
Foot and ankle exercises				
Introduction to foot care checklist				

**Table 4 healthcare-10-01143-t004:** Intervention, Content, Implementation and Outcomes.

Intervention	Content	Implementation	Outcome, Effect and Conclusion	Reference
Foot care programme	The foot care programme included: - Lower-extremity exam; - Foot care and footwear instructions; - Nail and callus care instructions; - Guidance on how to maintain a healthy lifestyle.	- A nurse specialising in wound and foot care inspects the feet. - Recommendations on how to wear proper shoes, stay hydrated, monitor calluses and ulcers, and live a healthy lifestyle is provided. - Patients with ulcers are sent to orthopaedists, vascular surgeons, infectious disease experts and wound care specialists. - Custom soles and orthoses are prescribed as needed.	- Patients Foot Care Behaviour (+) The foot care education was effective in reducing the frequency of peripheral neuropathy, and the absence of dorsalis pedis and posterior tibial arterial pulses, as well as enhancing the adequacy of footwear. No significant change was found in reducing the frequency of amputations, ulcers and Charcot foot.	Reda et al. [26]
Foot care education session	The foot care programme included: - foot assessment; - individual foot care education; - group foot care education; - special shoes. The control group did not receive the intervention.	- Foot assessment, individual foot care education, group foot care education, special shoes or inserts for each person, handouts.	- Patient Foot Care Knowledge (+) - Patient Foot Care Behaviour (+) The nephrology nurse has an opportunity to play a role in early detection, assessment and intervention for ESKD clients who have insensate lower extremities.	Neil et al. [27]
Education programme	The foot care programme included: - Importance of foot care for patients; - Foot examination (by a nurse, podiatrist and doctor); - Clarification of referral processes to specialist services.	- A single training session was provided by an experienced diabetes podiatrist to six specified nurses. - The training nurses delivered the information to their colleagues. - Monthly foot examination.	- Foot care behaviour (+) - Number of foot examinations (+) - Referral rate (+) A single education session can improve routine feet checks on diabetes patients receiving haemodialysis. NAFF administration has been linked to improved self-reported foot care behaviour, reflecting increased risk awareness in this population.	Brand et al. [28]
Education programme	The foot care programme included: - Foot examination (by training haemodialysis nursing staff); - Patient education on foot care; - Teaching nurses about the referral processes for specialist and foot care services in proportion to the patient’s needs for a referral.	- The intervention was a single education session programme for nurses that an experienced diabetes podiatrist conducted.	- Foot care behaviour (+) - Number of foot examinations (+) - Referral rate (+) The education programme elicited a change in the frequency with which nurses checked patients’ feet, supporting the belief that fast and simple foot checks would be most acceptable to nurses. - Clarification of referral pathways to podiatry as part of the nursing education programme enabled patients to be referred to this service.	Brand et al. [29]

## Data Availability

Not applicable.

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
