# Peer review of "Health Education Programmes to Improve Foot Self-Care Knowledge and Behaviour among Older People with End-Stage Kidney Disease (ESKD) Receiving Haemodialysis (A Systematic Review)"

_healthcare, 2022, doi:10.3390/healthcare10061143_

Round 1

Reviewer 1 Report

Alshammari et. al., in their review provided a very novel and interesting foot care educational programme for ESKD patients and provided evidence that foot care knowledge and self-care behaviors can be influenced in both diabetic patients receiving haemodialysis and healthcare professionals. Further, the various foot care education approaches identified in this review can be very beneficial in improving the patient care.

I only have few minor comments to make:

1.      Table 1 is not mentioned in the text. The table should be first introduced and then placed.

2.      Table 4 should be placed after text of 3.3.1 Programme structure and delivery characteristics

3.      Table 4 quality is poor and is unclear. The column headings (Format of Delivery) like Pho, Pam, Lect, De etc are not clear. Need to add more explanation about the programme in the text. Table 4 should be placed after text of 3.3.1 Programme structure and delivery characteristics.

4.      On page 17, line 657 ESRD should be corrected to ESKD

5.      The authors should include the shortcomings/hurdles there might be in applying the foot therapy in ESKD patients and how it varies from older people to younger people undergoing the treatment.

6.      Recent studies have identified the importance of  foot reflexology and back massages in improving sleep quality and fatigue in hemodialysis patients  The authors are suggested to include those studies and references.

7.      Several similar studies are available in literature such as Kaminski et al., Journal of Foot and Ankle Research, 2015. How is the current study different from them? What new insights are presented in the current study?

Reviewer 2 Report

Dear Authors,

Overall good job done on compiling the literature on this particular literature. Good job in bringing up critical area of patient education that will not benefit Kidney disease patients but also the diabetic population. I have few suggestions:

1) Tables can be briefed further as there is too much text.

2) Any relation between chronic kidney disease, foot care and diabetes- any positive outcome or extension of life or quality of life after intervention?

Reviewer 3 Report

In this review, Alshammari L et al, have provide a comprehensive review of studies which evaluated a correlation between health education programs relating to foot care and incidence of foot diseases like ulceration in elderly patients receiving hemodialysis for end-stage kidney disease (ESKD). Based on the review the authors were able to identify various factors that potentially help in the implementation of foot care programs for ESKD patients undergoing hemodialysis. The authors identified diabetes as a co-morbid condition in these patients and the studies reviewed were specific to patients that had diabetes as a co-morbidity in addition to ESKD. Overall, the researchers concluded that healthcare professionals/ practitioners need to develop and implement early assessment tools, and provide education to patients/ care-givers to prevent foot ulceration and/ or amputations in patients with diabetes and ESKD undergoing hemodialysis. The review is quite comprehensive and well written. Some of the minor concerns are enlisted below:

·         In abstract (line 22) please mention ESKD in all capital alphabets.

·         Did the authors find any correlation between the severity and/ or control of diabetes and the risk of foot ulceration and/ or amputations in ESKD patients undergoing hemodialysis? It seems unclear from the article and would be important to address this in the discussion section.

·         In figure 1 (flow diagram) it is unclear what N vs. n means. Please elaborate that in the figure legends.

·         In section 3.3.2, the authors start each paragraph where they discuss/ summarize the studies with the reference number. This style of presentation is awkward to read. Please write as ABC et al [reference number] showed…….

·         In table 4, under format of delivery the words are not complete. If abbreviations are used then please expand as a foot note in the table or re-format the table so that the words are readable.
